# Advancing In Vitro Tools for Oncologic Research in Cats and Dogs

**DOI:** 10.3390/vetsci12090815

**Published:** 2025-08-26

**Authors:** Chang He, Sven Rottenberg

**Affiliations:** 1Institute of Animal Pathology, Vetsuisse Faculty, University of Bern, 3012 Bern, Switzerland; chang.he@unibe.ch; 2Graduate School for Cellular and Biomedical Sciences, University of Bern, 3012 Bern, Switzerland; 3Bern Center for Precision Medicine and Cancer Therapy Resistance Cluster, Department for BioMedical Research, University of Bern, 3008 Bern, Switzerland

**Keywords:** in vitro model, comparative oncology, translational medicine, dog, cat, mammary tumor, cancer biology, 3D culture systems

## Abstract

Technological advances now allow scientists to grow cells in a dish outside the body, which has become a fundamental tool in life sciences and especially in cancer research. In recent years, there has been growing interest in using these technologies for pets such as cats and dogs. Both cats and dogs develop cancer similar to humans, such as mammary tumors (breast cancer), which can lead to devastating outcomes and make this disease highly important to study. Growing tumors in a dish makes it possible to study cancer cells without harming the patient, and, in the best case, artificially grown cells can help predict whether a drug will work. While these technologies are constantly improving, such that cells can even form three-dimensional mini tumors, we still face challenges in fully reproducing disease behavior in the body. This review summarizes the current state of tumor culture, with a focus on mammary tumors in cats and dogs.

## 1. Introduction

A cornerstone of both veterinary and human cancer research has been the development of in vitro culture systems, making the study of cancer cells outside of the patient possible. The first attempts at cell culture date back to the late 19th century, when a wide range of cell types were extracted from various organisms and kept viable outside of the donor for a limited amount of time. A pivotal discovery marked the establishment of the first immortalized human cancer cell line, HeLa cells [1,2]. The ability of cells to virtually proliferate indefinitely in a dish ex vivo has opened numerous avenues of biological research, particularly in cancer research. Over time, the repertoire of culturable mammalian cell lines has expanded substantially, and the culturing techniques have dramatically evolved with the addition of new technologies and deeper understanding of cellular biology. Nowadays, in vitro systems have become progressively more elaborate, attempting to emulate in vivo settings, including complex three-dimensional (3D) systems, from organoid culture to 3D bioprinting [3,4,5,6,7,8]. However, despite the advancement of in vitro culture systems, the complexity of in vivo animal models such as mice remains unrivaled, and their use still holds place in most preclinical studies [7,8,9].

Human and veterinary oncology are traditionally viewed as entirely different fields because of the differences between species. However, accumulating evidence has revealed striking parallels across species, making them increasingly difficult to distinguish from each other when inspected in detail. Similarly to humans, neoplastic diseases account for a significant proportion of morbidity and mortality in cats and dogs, according to records throughout America and Europe, with mammary tumors, cutaneous epithelial tumors, and lymphomas being frequently reported among the malignant cases [10,11,12,13,14,15,16,17,18,19]. In this context, both canine and feline mammary tumors (CMT and FMT) display striking similarities to human breast cancer (HBC), with both closely recapitulating key morphologic and clinicopathologic characteristics of their human counterparts [20,21,22,23]. CMT is a heterogeneous disease encompassing a variety of histological subtypes, and approximately half of all cases are malignant [24,25,26]. In contrast, FMT is typically highly aggressive and presents a high metastatic rate at diagnosis. Notably, FMT closely resembles the triple-negative subtype in human breast cancer (HBC)—defined by the absence of estrogen receptors (ERs), progesterone receptors (PRs), and human epidermal growth factor 2 (HER2) [21,25].

Furthermore, companion animals share a similar living environment with their owners and are therefore exposed to similar environmental risk factors. These parallels have led to the introduction of therapeutic strategies originally developed for humans into veterinary medicine. At the same time, cancer in pets offers significant potential to model human disease and serves as a valuable model to study tumor biology and treatment strategies. This reciprocal relationship between companion animals and humans is underscored by the One Health approach, which promotes the advancement of cancer research for both pets and owners [27,28,29,30,31,32]. For instance, the receptor tyrosine kinase inhibitor toceranib (Palladia ^TM^), initially developed for treating mast cell tumors in dogs, provided essential preclinical data for the development of its human analog, sunitinib (Sutent ^TM^), used for a variety of human cancer types [33,34]. Clinical trials involving toceranib not only improved treatment outcomes for dogs but also offered valuable insights into human cancer therapies. This example illustrates the potential in bridging the gaps in oncologic research between species and underscores the necessity for collaborative research efforts.

This review presents an overview of the application of in vitro culture systems, with a focus on canine and feline mammary tumors. We will explore both classical and advanced systems used to study cancer in cats and dogs. In addition, we will discuss the advantages and limitations of the currently available systems, highlighting the potential of comparative oncology bridging cancer research across species.

## 2. Conventional Two-Dimensional (2D) Culture

### 2.1. Overview of 2D Culture

To understand the advancements in in vitro systems, it is essential to examine the conventional two-dimensional (2D) culture first. The basic but still canonically used in vitro culture system involves propagating adherent cells in flasks as a monolayer supplemented with various formulations of growth media. This popular system is characterized by its low cost, user-friendliness, and scalability compared to the more complex three-dimensional (3D) systems discussed later. Cells are usually seeded, allowed to grow until reaching confluency, and passaged for continuous growth.

### 2.2. Immortalization

Before focusing on cancer-derived in vitro models, it is important to clarify that the classification of culturable cells as either “normal” or “cancerous” is an oversimplification. In most research settings, the ability to proliferate virtually indefinitely is essential, which poses a major limitation in culturing normal, non-transformed cells. In an ideal world, autologous normal cells—e.g., normal mammary cells derived from the same individual as the mammary tumor—would serve as corresponding controls.

However, even under optimal conditions, culturing primary normal somatic cells—those derived from normal donor tissues—will result in replicative senescence. This is primarily due to the activation of tumor suppressor genes, such as *TP53* and *RB1*, in response to the programmed shortening of telomeric DNA upon reaching a critical length. This occurs because DNA polymerase is unable to fully copy the telomeric ends during replication [35,36]. Typically, genetic modifications are required to counteract the telomeric shortening and/or bypass the tumor suppressor pathways. This can be achieved by expression of telomerase reverse transcriptase (TERT), a ribonucleoprotein which adds hexameric repeats during DNA replication. In somatic cells, TERT is physiologically repressed and is typically restricted to specific cell types such as germ cells, stem cells and activated lymphocytes, and the deactivation of tumor suppressor genes can result in an immortalized phenotype [37].

Most cancer cells typically bypass senescence by acquiring mutations affecting telomere regulation, and such alterations also need to be introduced in normal 2D cells to avoid senescence. A systematic analysis of telomere length in over 6800 human cancer samples revealed that 72% harbored TERT-expressing mutations, followed by samples showing alterations in *ATRX* or *DAXX*, key genes in the alternative lengthening of telomeres (ALT) pathway. In the remaining samples, telomere length was positively correlated with alterations in tumor suppressor genes such as *TP53* and *RB1* [36]. Hence, the typical immortalization protocols involve the introduction of a TERT-expressing vector and/or deactivation of *TP53* and *RB1*, usually by using viral oncogenes such as *Simian virus* 40 (SV40) or *Human papilloma* virus 16 (HPV16) [38].

Although extensive characterization of telomeric biology in companion animals remains elusive, various early studies identified frequent mutations in similar genes across tumors, including *TP53*, *RB1*, *ATRX*, and *TERT* [13,39,40,41,42]. These findings suggest that telomeric biology is highly conserved across species, extending beyond humans. Further evidence is provided by the use of human TERT-expressing vectors and/or human oncoviruses to successfully immortalize various normal canine and feline cells, suggestive of a shared biological foundation across species [42,43,44,45,46,47,48,49,50]. Despite these parallels, telomerase activity is rarely measured in companion animal tumor studies, which may lead to the underestimation of its prevalence. However, Uyama et al. [51] detected telomerase activity in all eight FMT cell lines established in their laboratory showing continuous growth—further supporting the cross-species homology of these mechanisms.

Notably, in mice, low levels of TERT are also expressed in adult somatic cells [52,53]. This special trait explains why mouse cells have significantly longer telomers (40–80 kb) [54] compared to humans (5–15 kb) [54], cats (5–26 kb) [55,56], and dogs (11–27 kb) [57]. Despite somatic mouse cells expressing TERT, the stricter regulation prevents indefinite telomere lengthening, contrary to immortalized cell lines [58,59]. Still, the fact that immortalization is easier achieved in mouse cells simplifies the process of establishing stable and long-term 2D cultures from mouse tissues.

### 2.3. Oncogenic Transformation

Given the overlapping features observed in vitro, it is crucial not to confuse immortalized cells with cancer cells. Immortalization only involves bypassing of replicative senescence, allowing cells to divide indefinitely. However, immortalized non-cancerous cell lines still lack certain properties exclusive to cancer cells, such as evasion of growth suppression or resistance to programmed cell death. For instance, normal cells enter a proliferative quiescence when they reach confluency as a monolayer known as contact inhibition or initiate apoptosis upon loss of anchorage through a process called anoikis. In contrast, cancer cells would continue to divide uncontrollably under these conditions [60,61]. In some cases, spontaneous immortalization in vitro may occur, a phenomenon frequently observed in rodent cell lines, likely due to the inherent TERT expression in these species (Figure 1) [58,62,63]. This phenotype was also shown in canine embryonic fibroblasts [42]. However, to the authors’ knowledge, this instance has not been reported in normal adult somatic cell lines from both dogs and cats, further underscoring the importance of telomerase activity in this process.

The soft agar assay is widely used to assess oncogenic transformation, where the ability to proliferate independently of anchorage is demonstrated, adherent cells being suspended in low-concentration agar, allowing the assessment of resistance to anoikis. This technique can also be adapted to culture some cancer lines in 3D, as will be discussed later. Another well-established method examines the tumorigenic potential of transformed cells in vivo using mice as xenotransplantation models. These models are considered more reliable as they reflect the interaction between the inoculated cells with the complex in vivo environment. Additionally, xenotransplantation models are useful to study long-term tumor progression and metastasis, with tumors often passaged in these mice to maintain tumor lineage and assess metastatic potential [64]. However, these models also have limitations, including the low engraftment rates, the foreign species origin of the microenvironment, the use of immunocompromised animals, the high cost, and ethical concerns [65].

Given this context, for immortalized cell lines to acquire typical hallmarks of cancer, additional pathway perturbations are required. Interestingly, species vary in the number of mutational hits required for successful oncogenic transformation. For instance, while human fibroblasts require perturbations in five different pathways—specifically p53, pRb, PP2A, telomerase, and Ras pathways—to achieve oncogenic transformation, mouse fibroblasts only require perturbations in p53 and Ras [66,67,68]. Such a disparity may explain the phenomenon called Peto’s paradox, where larger animal species do not show higher cancer incidences than smaller species, despite the significantly higher absolute number of cells [68,69]. This observation suggests that larger species harbor additional mechanisms that prevent an increase in malignant cell transformation, possibly correlating with the number of mutational hits required to transform cells across species.

Even more intriguing, studies on canine cell lines suggest that dogs resemble mice in their susceptibility to oncogenic transformation, despite the significant difference in body mass and life span. This was supported by the successful oncogenic transformation of embryonic immortalized canine fibroblasts using h-Ras in addition to the introduction of a catalytic telomerase subunit [42]. In another study [70], *TP53* knockout in canine fibroblasts with a presumptive mutation in *TFE3* was found to induce a tumorigenic phenotype. However, these results contrast with findings from a study [71] that used human cytomegalovirus (HCMV), known to induce oncogenic transformation in human cell lines [72]. While canine kidney cells were immortalized upon infection with SV40 or HCMV and anchorage-independent growth was promoted, only 1 out of 40 mice developed tumors when inoculated with these cells.

In contrast, the literature is significantly sparser regarding oncogenic transformation in feline cell lines. While successful immortalization was demonstrated in feline endothelial cells [50], respiratory epithelia [43], and intestinal epithelia [49] using combinations of SV40, HPV16, and hTERT, no investigations assessing their tumorigenicity have been conducted. Notably, the two studies involving epithelial cells [43,49] both observed that transformed cells acquired double positivity for cytokeratin and vimentin, suggestive of epithelial–mesenchymal transition (EMT). While EMT is a physiological process during embryologic transdifferentiation, EMT-inducing transcription factors are commonly upregulated in tumors and promote malignant transformation in mouse mammary epithelia [73]. Thus, the EMT phenotype in these feline epithelial cell lines may reflect potential malignant transformation.

Together, these findings highlight the complexity of tumorigenesis across species and the challenges associated with 2D cultures. They emphasize the need for future investigations in companion animals, possibly also by improving current cell culture systems. Elucidating the underlying mechanisms of the differences and similarities may result in more relevant translation across veterinary and human oncology.

## 3. Three-Dimensional (3D) Culture

### 3.1. Overview of 3D Culture

As technology advances, in vitro culture systems have become increasingly more sophisticated to bridge the gaps between in vitro models and their primary counterparts. While conventional 2D culture systems provide powerful ex vivo tools, the loss of native multicellular architecture and change in cell shape through the growth on hard plastic can substantially impact gene expression [74], resulting in altered cell differentiation [75] and metabolic activity [76]. The limitations of 2D cultures drove the development of more physiologically relevant in vitro three-dimensional (3D) models, such as organoids, tumoroids, and spheroids, using various systems aiming to reflect the in vivo tissue environment. These 3D systems can be broadly divided into scaffold-based or scaffold-free approaches, both aiming to recapitulate in vivo complexities.

In this context, scaffolds refer to supportive materials that mimic the native extracellular matrix (ECM) for cells to grow on or within. Alternatively, scaffold-free approaches, including static suspension cultures, dynamic microfluidic devices, and bioreactors, rely on meticulously controlled biochemical cues to promote 3D growth [77]. With the wide range of available 3D culture systems, determining the most suitable one depends on the experimenters’ specific needs. Scaffold-based systems excel in mimicking in vivo microenvironments but require cell-type-specific optimization. While they are limited in nutrient access and waste removal due to restricted diffusion, scaffold-free systems offer high-density expansion and superior control over the culture environment. On the other hand, scaffold-free systems lack the mechanical support and are often more costly to implement in term of consumables and equipment [78]. Both approaches rely on specific culture medium formulations that vary dramatically across studies, significantly impacting cell behavior and experimental outcomes, making the selection of an appropriate medium formulation critical [77,79,80]. Given the vastness of the topic of 3D culture technologies, we will narrow our discussion to the application of 3D models to highlight their significance in advancing veterinary oncology.

### 3.2. Organoids vs. Spheroids vs. Tumoroids: Definition Matters

To better understand the distinct advantages of 3D models, it is important to clarify key terminologies first. Across studies, authors classify their in vitro models as organoids, spheroids, and, in the context of the mammary gland, mammospheres. Etymologically speaking, a spheroid (“sphere-like”) describes a geometric feature—likewise, a mammosphere refers to a spherical structure derived from the mammary gland—whereas an organoid (“organ-like”) implies organotypic complexity. This interchangeable use of these terms to describe various types of 3D models has been a source of confusion. Essentially, the core feature of organoids is their ability to arise from pluripotent or multipotent stem cells and to self-renew into multicellular 3D structures, mimicking key morphological and functional characteristics of an organ [4,80,81]. This process often depends on Wnt signaling, a critical pathway promoting the expansion of stem cells and organoid formation [77,80]. Notably, Wnt signaling activation also induces TERT expression, contributing to the long-term culture capability of organoids [82].

Given this context, organoid culture relies on the natural differentiation pathways of these cells—meaning that the composition and architecture of the final organoid depend on both the commitment state of the starting stem cells and the culture conditions (Figure 2). For instance, naïve adult stem cells from the intestinal crypt will differentiate in various epithelial portions of the intestine. This applies similarly to the mammary gland. On the other hand, genetic modification can reintroduce the required stemness in differentiated somatic cells, allowing for the formation of complex brain organoids from a single induced pluripotent stem cell (iPSC) [77,80,81,83].

While the definition of an organoid is generally straightforward, classifying tumor-derived 3D structures may pose a challenge, particularly regarding solid tumors that lack clear morphological heterogeneity and architecture. Traditionally, “spheroids” refers to 3D structures generated from any cells forming simple, miniature spheres. One of the earliest models, developed in the 1970s, used hamster cancer cell lines cultured in spinning flasks to keep cells in suspension [84]. Although spheroids lack a tissue-specific architecture, they exhibit a characteristic layering due to the centripetally decreasing nourishment gradient, reflecting the behavior of poorly vascularized solid tumors and provide physiological relevance to drug response studies similar to organoids. Building on this model, researchers typically refer to spherical 3D structures derived from tumor cell lines as spheroids [85,86], implying that these structures have low heterogeneity and complexity due to the observed phenotypes in 2D cultures. While convenient, this nomenclature may overlook the potential organotypic complexity of these 3D tumor models. For instance, Petersen et al. [87] showed that 2D cell lines of non-cancerous mammary epithelium cells formed an organotypic glandular architecture with various forms of cellular differentiation when 3D culture conditions were introduced—illustrating the potential for cells to generate organotypic complexities, even after being adapted to 2D growth.

In this context, 3D culture studies of CMT exhibit organoid characteristics, displaying glandular histomorphological features while maintaining the mutational landscape and immunophenotypic signature of the original tumor tissue [88]. In contrast, such organoid complexities may be less evident in 3D models derived from tumors with rather compact solid growth, such as canine urothelial tumors [89].

The distinction between tumor organoids and tumor spheroids is crucial, as the way we conceptualize tumor-derived 3D structures depends on the tumor growth model we subscribe to. Therefore, we propose the term “tumoroids” (“tumor-like”) as a more appropriate nomenclature for tumor-derived 3D models. According to the cancer stem cell (CSC) theory, only a subset of specialized cells within a tumor are responsible for disease initiation and driving disease progression. These CSCs can additionally differentiate into a variety of non-tumorigenic cells with a limited capacity for self-renewal. Given this context, in vitro conditions suitable for organoid culture should logically produce tumor organoids from these CSCs. However, some researchers may argue that the non-tumorigenic population in 3D HBC cultures may result from contamination by surrounding untransformed cells [90]. It is an interesting paradox that organoid culture conditions favor the growth of normal tissues over cancerous ones [91].

Although the CSC model is still debated, there are cancer cell subpopulations expressing stem cell markers such as CD133 or CD44 [92,93]. Both markers have also been detected in 3D CMT culture via Western blotting, although the bands appeared faint [94]. Along these lines, the authors performed lineage tracing experiments to track individual cell fate over time, combined with ALDH activity measurements providing evidence for a hierarchical structure within these CMT tumor organoids [79,94].

In contrast, the stochastic model of carcinogenesis posits that any oncogenetically transformed cells can expand clonally and equally contribute to tumor growth [95]. This model suggests a tumor architecture, where all transformed cells drive tumor progression without the need for hierarchical differentiation. There may still be intra-tumoral heterogeneity, but which subpopulation thrives is stochastic and not the result of a pre-defined program. In case the tumor cells are fairly homogeneous, tumor spheroids may better align with the idea of a homogenous cell aggregation of limited complexity.

At the same time, the line between both models starts to blur given the concept of cancer cell plasticity, positing a continuous transition of cancer cells into—or their reversion to—progenitor-like states [61]. And if there is plasticity, can a spheroid become a tumoroid and vice versa? Does this mean that the distinction is ultimately not as rigid as we thought and that one should rather focus on their cellular origin—whether derived from cancerous cells (i.e., tumoroids) or non-transformed cells.

Ultimately, without clear evidence of differentiation or organization, the distinction between a tumor organoid and a tumor spheroid may require additional investigation such as lineage tracing experiments. While organoid formation typically relies heavily on the culture medium supplements, cancer cells may lose these dependencies to activate oncogenic pathways [96], allowing the formation of 3D organoids even under suboptimal conditions. This further complicates the classification, suggesting that a more encompassing term, such as tumoroids (“tumor-like”), might be appropriate, with further subclassification based on whether a structure exhibits an organoid- or spheroid-like phenotype.

## 4. Mammary Tumor Models

### 4.1. Advancements and Challenges in Establishing In Vitro Models for Mammary Tumors

With the growing attention to alternative HBC models, there has been a substantial increase in the number of mammary tumor models in companion animals (Table 1). Significant progress has been made with 2D CMT and FMT models. However, the development of 3D tumoroids remains scarce yet potentially more impactful. This scarcity of 3D models may stem from challenges in optimizing growth medium formulations and financial constraints associated with the necessary reagents.

Efforts to develop 3D CMT models have seen some success. Cocola et al. [79,94] were among the first to create CMT tumoroids, using tissue samples of eight CMT patients. However, the actual success rate of producing the primary organoids remains unclear, as failed attempts may not have been reported. In addition, only short-term cultures lasting up to five passages were achieved. In contrast, Inglebert et al. [88] established the first large-scale organoid biobank, generating 23 tumoroid lines derived from 16 dogs (some dogs had multiple tumors), yielding a success rate of around 75%. These tumoroid lines demonstrated long-term potential, sustaining growth for up to 20 passages. Notably, these numbers are remarkably higher than those reported in HBC tumoroid cultures with success rates as low as 3.33% [90]. That said, the authors observed a reduction in estrogen receptor expression at the protein level with extended passages, suggesting some degree of cellular adaptation to the in vitro conditions.

Despite advancements in CMT models, the development of 3D FMT models remains elusive. Success rates for establishing cell lines have ranged from 2.5% to 40% [51,97,98]. These relatively low numbers could indicate potentially inherent challenges in establishing primary feline tumoroids, possibly attributed to suboptimal culture conditions. Additionally, the actual number of failed attempts often remains unclear due to the tendency for negative results to go unpublished, further complicating progress in this field. Despite these apparent challenges, the existence of feline intestinal and liver organoids [99,100,101] established using similar protocols to those for canine and human organoids suggests that the development of a 3D FMT culture may be feasible. Refining protocols from CMT research could help address the obstacles to establishing FMT tumoroid models.

**Table 1 vetsci-12-00815-t001:** Summary of studies on establishing in vitro canine and feline mammary tumor models.

Species	Model Type	Tumor Source	Number of Animals	Reference	Highlights
Canine	2D	Primary carcinoma and metastases	2	Van der Burg, 1989 [102]	4 tumor cell lines established
		Primary mixed adenoma	1	Priosoeryanto et al., 1995 [103]	1 tumor cell line established
		Carcinoma and adenoma	5	Hellmén, 1992 [104]	5 tumor cell lines established Using 3 different modes of sampling: 1) tumor fragment DMSO frozen vs. 2) fine-needle aspirate vs. 3) tumor fragment freshly processed
		Primary carcinoma and metastases	4	Uyama et al., 2006 [105]	4 pairs of tumor cell lines from 4 dogs (primary and metastatic) established
		Primary carcinoma	1	Caceres et al., 2015 [106]	1 tumor cell line established
		Primary carcinoma	1	Mei et al., 2021 [107]	1 tumor cell line established
		Primary mixed adenocarcinoma	1	Li et al., 2021 [108]	1 tumor cell line established
		Included tumor types: complex carcinoma, complex adenoma, and mixed adenoma	10 out of 12	Yeom et al., 2023 [109]	10 tumor cell lines established Targeted sequencing for two variants of PIK3CA Validated for in vitro drug assays
		Included tumor types: complex carcinoma, mixed tumor, in situ carcinoma, and simple carcinoma	8	Park et al., 2024 [110]	8 tumor cell lines established Targeted sequencing for a variant of PIK3CA Validated for in vitro drug assays Identified over 200 highly expressed genes and enriched EMT signatures
Canine	3D	Not specified	8	Cocola et al., 2009 [94]	8 tumoroid lines and 8 normal mammary glands established Could be maintained for up to 5 passages
		Included tumor types: complex carcinoma, simple carcinoma, mixed tumor, mixed carcinoma, and malignant myoepithelioma	16	Inglebert et al., 2022 [88]	23 different tumoroids and normal mammary organoids out of 32 samples established Suitable for long-term culture Validated for gene editing and in vitro drug assays Verified conserved mutational landscape in in vitro models using whole-genome sequencing (WGS)
Feline	2D	Primary carcinoma	4 out of 30	Norval et al., 1985 [97]	4 tumor cell lines established
		1× primary carcinoma 3× pulmonary metastases	4 out of 135	Minke et al., 1991 [96]	4 tumor cell lines established
		Primary carcinoma and metastases Including 1 thoracocentesis sample	5 out of 13	Uyama et al., 2005 [51]	5 cell lines established Telomerase activity detected in all cell lines
		Primary carcinoma	1	Borges et al., 2016 [111]	1 cell line established
		Primary carcinoma	1	Granados-Soler et al., 2018 [112]	1 cell line established Immunophenotypic characterization Copy-number variation analysis of cell lines and tumors Validated for in vitro drug assays Investigated EMT-related gene expression levels

### 4.2. Predictive Drug Response and Biomarkers in Mammary Tumor Models

Building on the development of these models, one of the arguably most clinically relevant aspects of primary in vitro models is their potential to predict drug response without the ethical concerns of in vivo testing. Several studies have investigated drug response involving CMT and FMT models to identify clinically translatable vulnerabilities [88,109,110,112,113]. For example, as early as 1981, von Hoff et al. [114] demonstrated that human tumoroids grown in soft agar had an impressive 88% sensitivity and 94% specificity when subjected to a panel of cytotoxic drugs. This study highlights the remarkable potential of primary in vitro models to serve as functional tools for predicting therapy response. However, a limitation of this application was the low success rate of 25% in growing the samples in vitro, likely due to the suboptimal growth medium formulation. More recent attempts, for example, in colorectal cancer, the role model for developing tumoroids, the response of patients to oxaliplatin was predicted with a sensitivity of 70% and a specificity of 71% using patient-derived tumoroids [115]. This shows that we still need to substantially improve the current tumoroid growth conditions to achieve reliable predictions.

With the recent revelation of *PIK3CA* mutations being frequently observed in CMTs, researchers are focusing on exploiting this mutation for targeted therapy, similar to approaches in HBC [108,109,110,116]. Both 2D [109,110,116] and 3D [88] CMT lines have conserved their matching mutations in vitro. However, results across studies have been variable. While most studies [88,109,110] showed increased drug sensitivities associated with *PIK3CA* mutations, Maeda et al. [116] reported mixed results between *PIK3CA*-mutated lines and wild-type lines. Still, most of these studies used a targeted approach to identify mutated CMT lines, in which additional genomic alterations may not have been accounted for, which could potentially have impacted drug response. Although all authors used CMT models harboring the same mutations, the tumors may differ in their biological behavior. For instance, Yeom et al. [109] found no evident correlations between the mutation and tumor malignancy. They further observed cells reminiscent of chondrocytes in two of the CMT cell lines, further underscoring the heterogeneity between CMT lines. These findings highlight how even well-characterized mutations, such as *PIK3CA*, warrant further investigation and more standardized methods to better understand their impact on tumor behavior and drug response in CMT models.

In contrast, investigations into drug response in FMT models remain limited. Recent copy-number variation (CNV) analysis identified several altered cancer-related genomic regions as prognostic markers, supported by retrospective clinical survival data [117]. While there are several targetable genes within these altered genomic regions, including *JAK2*, *PD-L1*, and *PD-L2*, further validation experiments are needed to establish their suitability as appropriate candidates for targeted therapy approaches. Despite the potential application of immune checkpoint inhibitors targeting *PD-L1* and *PD-L2,* the current lack of established FMT in vitro co-culture systems with immune cells presents an additional obstacle. Limited studies have tested drug response in FMT models so far. One study [113] subjected an FMT cell line to 5-fluorouracil (5-FU), an antimetabolite for disrupting DNA synthesis, in combination with a panel of drugs known for their synergistic effects in HBC patients. While 5-FU significantly decreased cell viability, none of the combinations proved to have additional effect, contrary to HBC cell lines. The authors concluded that these results may reflect the aggressive biological behavior of the FMT cells used. The precise reasons for the lack of drug response in FMT remains unclear, as several factors are capable of influencing experimental outcomes, ranging from growth medium formulation to even passage number. For instance, it was shown that an FMT cell line’s sensitivity to doxorubicin doubled with successive passages, similar to findings in human cell lines [112,118]. One possible explanation could be the enrichment of certain cells through selective pressure, a phenomenon observed in HBC studies, where genomic aberrations shift with increased passaging [90]. In another study, Gameiro et al. [119] demonstrated a synergistic inhibitory effect of receptor tyrosine kinase inhibitors (lapatinib and neratinib) and rapamycin in FMT lines, including the same line used in the 5-FU study. However, passage numbers were not reported, which is common in many human and veterinary studies, making it difficult to assess the potentially confounding effects of passage stage on drug responses. These examples emphasize the need for a more stringent standardization across studies to improve experimental reproducibility and to derive meaningful results.

### 4.3. Exploring Novel Biomarkers with CRISPR/Cas9 Screening

While significant attention has been given to drug testing and mutation-specific drug response prediction, genetic screenings using CRISPR/Cas9 technologies have recently emerged as a powerful tool for discovering novel predictive biomarkers, as well as investigating tumor-specific vulnerabilities and resistance mechanisms. In contrast to traditional genomic screening that identifies mutations in wild-type genomes, CRISPR/Cas9 screening provides functional context for genomic alterations. For example, CRISPR/Cas9 screening enabled the identification of *LRP6* in a CMT cell line as a critical tumor-specific host factor that promotes infections by the oncolytic virus, the Ondersteepoort strain of Canine Distemper Virus (CDV-OP) [120]. These findings lay the mechanistic framework for refining oncolytic viral therapies across different tumor types and species, potentially enhancing their precision and therapeutic effectiveness.

Moreover, CRISPR/Cas9 screening was successfully applied to both CMT tumoroids and normal canine mammary organoids using a custom-designed CRISPR/Cas9 sub-library targeting over 800 druggable genes. This approach opens new opportunities for uncovering cancer-specific vulnerabilities and mechanisms [88,121]. A key advantage of this approach is the ability to directly compare normal and tumor cells from the same animal, deepening our understanding of tumor-specific genetic dependencies to further improve cancer treatment and the development of novel therapies. Although these approaches have not yet been implemented in FMT models, their application to feline models appears promising.

Despite the pressing need for effective therapies in FMT, such advanced methodologies remain significantly underrepresented in FMT research. FMT is often proposed as a model for triple-negative breast cancer (TNBC) in women. While TNBC is recognized as the most aggressive subtype of HBC, it is frequently susceptible to poly (ADP-ribose) polymerase inhibitors (PARPi) due to the homologous recombination deficiencies (HRDs) that arise from mutations in *BRCA1/2* genes [122]. Genetic studies investigating *BRCA1/2* mutations in cats have produced mixed results. One study [123] reported no mutations in a cohort of 24 patients, whereas a different study [124] claimed that 3 out of 9 cats carried variants with moderate impact. It is important to note that although *BRCA1/2* mutations lead to HRDs, there are also BRCA-independent genetic alterations that can result in HRDs. Considering the remarkable clinicopathologic resemblance between FMT and TBNC, it is plausible that FMT may exhibit HRDs as well. In vitro FMT models present promising ex vivo investigations to explore this hypothesis by establishing functional testing for HRDs in FMT [125].

### 4.4. Lost in Translation

Despite progress in modeling mammary tumors in vitro, translating some findings into clinical applications remains challenging, even using 3D tumoroid models. Current scaffold-based CMT tumoroids can mimic certain aspects of the in vivo environment, but they are still largely limited to epithelial monocultures, excluding stromal and immune cells that are crucial in shaping tumor behavior. This limitation likely contributes to the observed discrepancies in drug sensitivities between in vitro and in vivo mouse mammary tumor models, due to the protective effects of the tumor microenvironment [126,127]. For instance, studies in canine mast cell tumors have demonstrated that co-culturing with stromal cells significantly improves cell viability [128]. Similarly, in a study with human cell lines [129], cancer-associated fibroblasts (CAFs) were found to secrete a specific profile of growth factors and chemokines, promoting proliferation as well as inducing resistance to targeted antibody therapy.

These observations raise the important question of how accurately in vitro systems can predict clinically relevant drug efficacies in cancer therapy. In this context, the use of patient-derived tumoroid cultures to guide individualized treatment decisions conceptually aligns with bacteriologic culture-based diagnostics. Interestingly, both in vitro tumoroid culture and antimicrobial susceptibility testing (AST) share similar limitations regarding their predictive value in vivo. In both cases, the systems tend to be selected for specific subpopulations and are susceptible to pre-analytical errors, including variability in clinical sampling and processing. While AST generally correlates well with clinical outcomes, therapy failure still occurs due to factors such as biofilm formation or drug-limiting conditions at the infection site (e.g., pH or perfusion), similar to tumoroid models without a tumor microenvironment [130,131].

The profound influence of the tumor microenvironment on in vitro models emphasizes the need for comprehensive characterization and optimization of co-culture systems in the future. Accurately replicating the tumor microenvironment requires detailed profiling of the various cellular components involved. Advanced transcriptomic approaches, such as single-cell RNA sequencing (scRNAseq), provide powerful tools for extensive mapping of the cellular composition and gene expression profiles within both the tumor and its microenvironment [132]. Moving forward, integrating these approaches will be pivotal for improving the fidelity and translational relevance of in vitro models. Nonetheless, financial constraints pose a persistent barrier. The high costs associated with essential culture reagents and advanced technologies hinder the development and optimization of these complex in vitro systems, ultimately dictating the pace of progress in veterinary oncology.

In summary, we think that the challenges summarized in Figure 3 need to be addressed to improve the predictivity of 3D tumoroids in mammary tumors.

(a)Reproducibility and Standardization Issues

Variability in culture conditions remains a major obstacle. Mammary tumoroid protocols, whether derived from human or canine tissues, are frequently based on “homebrew” media compositions that vary between laboratories. Even when commercial kits like those adapted for veterinary oncology are used, slight differences in growth factors, extracellular matrix components, and culture duration may result in significant variability in drug sensitivity readouts. This variability complicates the predictive accuracy when translating in vitro findings to clinical or veterinary settings.

(b)Incomplete Recapitulation of the Native TME

Although 3D tumoroids have provided an improved platform over conventional 2D cultures, current models for breast and mammary tumors often suffer from a reduction or loss of non-epithelial components. For instance, while human breast cancer tumoroids may retain certain epithelial characteristics, they often lack a fully representative TME—missing stromal support cells, resident immune cells, and the complex extracellular matrix—which are essential for accurate drug response prediction. Moreover, hormone receptor-positive tumor cells appear to lose these during the tumoroid culture. These components may lead to underestimation of therapeutic resistance mechanisms typically observed in vivo.

(c)Translational Gaps and Predictive Accuracy

Similar to findings in CRC, the sensitivity and specificity of 3D tumoroid models in predicting drug responses in breast and mammary tumors remain moderate. This highlights the discrepancies between in vitro drug responses and the complexity of tumor behavior in a living organism. These discrepancies are partly attributed to the heterogeneous nature of both human and canine mammary tumors, where distinct subtypes (e.g., triple-negative breast cancer in humans and complex carcinoma in dogs) present unique challenges for in vitro modeling.

(d)Addressing Tumor Heterogeneity

Intrinsic heterogeneity within breast tumors is a well-documented phenomenon, affecting prognosis and therapeutic responsiveness. Despite maintaining some level of histological and molecular fidelity, current mammary tumoroids do not fully capture inter- and intra-tumoral diversity. Incorporating methods such as multi-regional sampling and single-cell sequencing into tumoroid platforms may improve their predictive utility, allowing for a more tailored approach—both in human oncology and in veterinary practice.

(e)Cost and Practical Limitations

Generating and maintaining mammary tumoroids is often more laborious and costly compared to traditional models. This is especially true when optimizing conditions to preserve aspects of the TME. While some studies on CRC have demonstrated cost-effective methods, the financial constraints in veterinary research may require further innovations, such as automation and high-throughput screening methods, to achieve more consistent and economically feasible results in mammary tumor models.

To enhance the translational applicability of mammary tumoroids, future research should focus on several key directions. First, integrating multi-cellular co-culture systems that incorporate fibroblasts, immune cells, and endothelial cells will better recapitulate the complexity of human and animal tumor microenvironments. Second, establishing standardized culture guidelines addressing growth media composition, ECM substrates and culture duration will be crucial to reduce inter-laboratory variability. Additionally, advances in automation and AI-assisted monitoring should be leveraged to generate more refined and predictive readouts. Finally, rigorous in vivo validation is critical to correlate tumoroid drug response profiles with clinical outcomes in both canine and feline mammary tumor cases.

## 5. Final Remarks

Overall, the effort to replicate in vivo conditions in vitro has led to substantial progress in the field of veterinary oncology for cats and dogs. While 2D models have established foundational knowledge, the development of 3D tumoroid models presents an exciting opportunity for more accurate experimental outcomes. However, critical challenges remain, especially in establishing robust FMT models, addressing experimental variation across studies, and integrating the tumor microenvironment in vitro. Ultimately, overcoming these challenges will be crucial for advancing the field of veterinary oncology, ensuring that cancer research yields meaningful experimental outcomes for companion animals.

## Figures and Tables

**Figure 1 vetsci-12-00815-f001:**
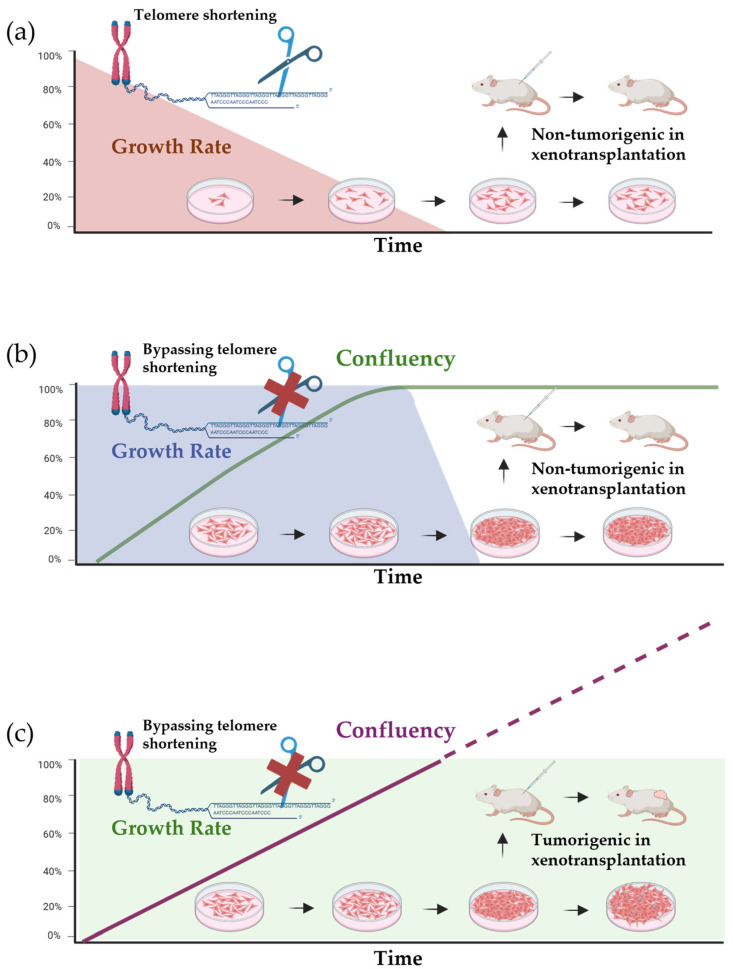
Comparison of cell behavior in conventional adherent 2D culture. (**a**) Normal primary cells undergo a limited number of cell divisions and enter replicative senescence due to telomere shortening. They are susceptible to contact inhibition and anoikis. Xenotransplantation does not result in tumor formation. (**b**) Non-tumorigenic immortalized cells bypass telomere shortening and replicative senescence but still respond to contact inhibition and anoikis. Xenotransplantation does not result in tumor formation. (**c**) Traditional cancer cell lines exhibit uncontrolled proliferation and anchorage-independent growth. Xenotransplantation results in tumor formation.

**Figure 2 vetsci-12-00815-f002:**
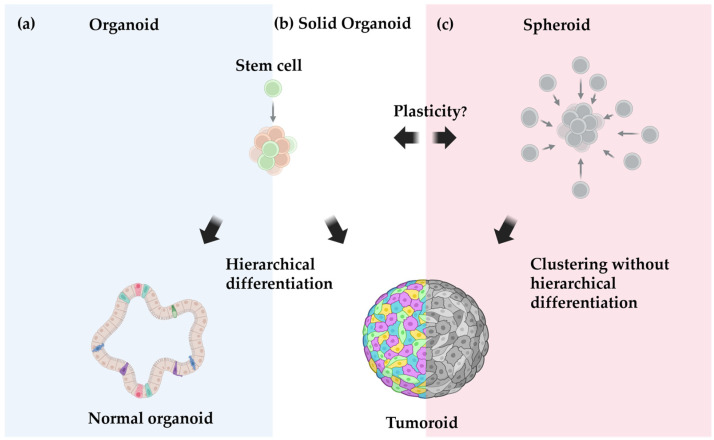
Conceptual comparison of organoid and spheroid models. (**a**) Normal organoids derived from stem cells exhibit hierarchical differentiation, organotypic architectures, and the capacity for self-renewal. (**b**) Tumor organoids derived from cancer cells with stem-like properties can also exhibit hierarchical organization, although this may be obscured in tumors with solid growth patterns. (**c**) Tumor spheroids consist of clonally expanding cancer cells that lack hierarchical organization. Cancer cell plasticity may enable transitions between tumor organoids (**b**) and tumor spheroids (**c**).

**Figure 3 vetsci-12-00815-f003:**
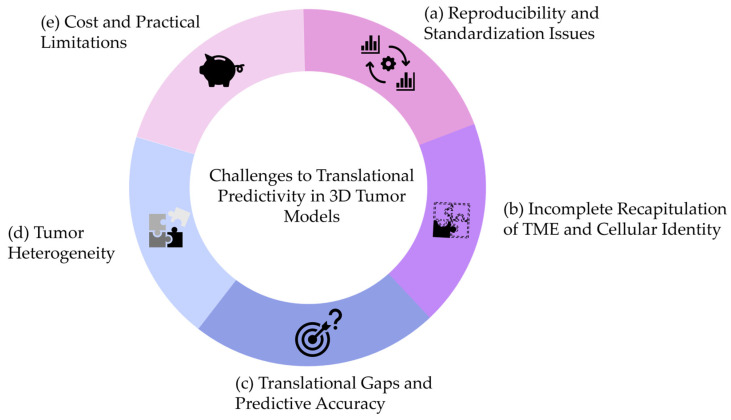
Summary of the challenges that need to be addressed to improve the predictivity of 3D tumoroids in mammary tumors.

## Data Availability

Not applicable.

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
