# Peer review of "Advancing In Vitro Tools for Oncologic Research in Cats and Dogs"

_vetsci, 2025, doi:10.3390/vetsci12090815_

Round 1

Reviewer 1 Report

Comments and Suggestions for Authors

The reviewed article addresses a relevant topic in the field of veterinary oncology, particularly in the context of research, by discussing widely used experimental tools worldwide. The manuscript is comprehensive, well-structured, and written in a fluent style, with only minor language-related issues.

Some improvements are recommended:

  1. Table 1 should be reformatted into a diagram and presented as a figure, which would improve clarity and visual impact.
  2. The section on the translational applicability of mammary tumoroids, currently presented in bullet points at the end of the review (immediately before the final considerations), would be more appropriately presented as a coherent paragraph.

The figures are appropriate to the technical content, well-structured, and adequately described in their respective captions.

Overall Recommendation: This is a quality review article. Acceptance is supported pending minor revisions.

Reviewer 2 Report

Comments and Suggestions for Authors

The manuscript adequately explores the in vitro tools for cancer research in dogs and cats, describing the advantages and disadvantages of the various alternatives. In my opinion, the text should be accepted for publication without modifications.

Author Response

We sincerely thank the reviewer for their positive evaluation and support for publication. While no modifications were requested, we have added a brief paragraph (lines 441–446) providing additional context on inter-laboratory differences in FMT drug response studies to further strengthen the manuscript. Additionally, we corrected several typos in Table 1 to improve accuracy and readability.

In another study, Gameiro et al. [122] demonstrated a synergistic inhibitory effect of receptor tyrosine kinase inhibitors (lapatinib and neratinib) and rapamycin in FMT-lines including the same line used in the 5-FU study. However, passage numbers were not reported, which is common in many human and veterinary studies, making it difficult to assess the potentially confounding effects of passage stage on drug responses.